# Automated Generation of Daily Evacuation Paths in 4D BIM

**Kyungki Kim [1] and Yong-Cheol Lee [2],***

[1]   Department of Construction Management, University of Houston, 4734 Calhoun Road #111, Houston, TX 77204-4020, USA; kkim38@central.uh.edu

[2]   Bert S. Turner Department of Construction Management, Louisiana State University, 3315E Patrick F. Taylor Hall, Baton Rouge, LA 70803, USA

*   Correspondence: yclee@lsu.edu; Tel.: +1-225-578-5483



**Featured Application: The proposed approach can potentially become one of path finding applications or add-ins of Building Information Modeling (BIM) tools. This application utilizes detailed daily or weekly construction plans to generate all possible evacuation paths of crews.**

**Abstract:** Spatial movements of workers and equipment should be carefully planned according to project plans. In particular, it is crucial for workers' safety to prepare emergency evacuation paths according to changing construction site configurations and construction progress. However, creating evacuation paths for all crews for each day can be an extremely labor-intensive task if it is done manually. Consequently, in most construction projects, evacuation plans are not provided to managers and crews throughout the entire construction. Even state-of-the-art technologies do not suggest ways to generate evacuation paths according to changing progresses presented in 4-Dimensional Building Information Model (4D BIM). This research proposes a framework to automatically analyze, generate, and visualize the evacuation paths of multiple crews in 4D BIM, considering construction activities and site conditions at the specific project schedule. This research develops a prototype that enables users to define parameters for pathfinding, such as workspaces, material storage areas, and temporary structures to automatically identify the accessible evacuation paths. This prototype shows the secured evacuation paths in the 4D BIM environment and allows the users to organize the automatically generated evacuation paths. A case study using the BIM model of a real construction project involved in this paper demonstrates the potential of the proposed method.

**Keywords:** BIM; safety; path planning; A-Star Searching; evacuation

## 1. Introduction

Throughout a dynamically changing construction project, managing limited spaces of a construction site is imperative to seamlessly facilitate linearly and cross-linking planned construction activities and operations. A project manager or an associated industry professional must carefully create a site logistic and safety plan that explicitly illustrate secured work spaces and accessible pathways according to construction phases and various moving paths for workers, equipment, and material delivery. In addition, before initiating daily work on a construction site, work crews in diverse domains must be trained and educated regarding the evacuation paths, emergency exits, and secured spaces so that they can rapidly escape in time when emergency situations occur. In a tool-box meeting where each work team shares daily work agendas, work crews are supposed to be educated about daily task details, designated work locations, relevant safety requirements, emergency manuals, and evacuation path information. However, such construction space management and path planning require labor-intensive

and time-consuming tasks because construction site conditions are generally varied by numerous factors, such as construction schedules, associated work locations, major equipment, temporary structures, and others. Especially, interactively creating and updating emergency evacuation plans according to the changing circumstances may require significant time and effort. Usable evacuation paths should be identified, secured, and shared with related work crews and construction managers through construction management tools or Building Information Modeling (BIM) software.

Despite this importance, there are several challenges in interactive evacuation path planning because of the complexity of modern construction projects and manual planning practices. A construction environment is generally formed by building objects (e.g., walls, columns, floors) and various non-building objects (e.g., workers, equipment, and temporary structures). Since the status and locations of these objects dynamically change even more often than on a daily basis, it is challenging for individual work crews or construction managers to perceive all the building and non-building changes in order to identify optimal evacuation paths. The lack of identification of secured pathways on site frequently results in additional cost growth and schedule delays during the construction phase. In particular, this unforeseeable condition and working environment of the construction site is one of the primary factors affecting workers' safety. When it comes to safety, it is challenging to predict construction accidents and emergency situations during construction. To provide against such possibilities, a well-developed emergency and evacuation plan representing the existing construction conditions is mandatory. Even though OSHA standards [29 CFR 1910.38(a)] entail the emergency action plan (EAP), the interactive evacuation plans during the construction phase still require further study. In current practices, construction site managers and domain professionals use their past experience and intuitive understanding about changing workplace conditions when planning moving paths of workers, logistics, and vehicles [1]. This relies on subjective judgment that can produce suboptimal or unsafe paths.

With the advancement of data representation and information technology, BIM has increasingly developed to fulfill the evolving demands of architecture, engineering, construction, and facility management (AEC-FM) industries. The remarkable gist of this BIM technology is the direct integration of a geometry and its corresponding information. This integration enables domain professionals to intuitively manage design and construction resources on the BIM platform and to virtually analyze the construction processes and operations in accordance with the given 3D model. One of the promising BIM applications is 4-Dimensional Building Information Model (4D BIM) that merges construction schedule information into a 3D BIM environment. 4D BIM is considered one of the most advanced ways to digitally represent, not only dynamically changing construction sites, but also constantly shifting construction working conditions through the construction timeline. In addition, since BIM generally entails resources and information of relevant objects and relationships, professionals in the construction industry are able to integrate indispensable construction data into one consolidated platform and manipulate them for distinct purposes during the construction phase. With its time-dependent and realistic visualization, various moving paths (e.g., evacuation, site entrance, material delivery) can be planned on top of the 4D BIM platform.

However, the benefits of current 4D BIM applications and analysis are still limited to the visualization of expected construction site conditions. Since 4D BIM can explicitly represent construction processes with associated resources based on a planned schedule, it is a desirable approach to plan work procedures, work crews' paths, and evacuation scenarios according to a planned schedule. However, it can be extremely labor-intensive for project managers to manually analyze complex 4D BIM to establish path plans for multiple work crews and different purposes (entrance, material delivery, evacuation) on a daily basis. Unfortunately, even though several 4D BIM applications have been proposed for daily safety hazard identification [2,3], path planning within 4D BIM has been overlooked.

From this perspective, there is a need for BIM-based automation that generates evacuation paths throughout the construction processes. This research presents a method to automatically generate rational evacuation paths of multiple crews using information in 4D BIM. The analyzed paths according

to dynamic site conditions and changing construction progress will be valuable assets for field managers and the workforce to secure guaranteed pathways and accessible spaces.

This paper is organized as follows. The literature review section presents a review of state-of-the-art approaches in the area of 4D BIM-based automated analysis and BIM-based construction path generation. A point of departure, a research objective, and a study scope are presented. The next section presents the framework and algorithms of the proposed evacuation path generation in 4D BIM. The case study section presents a validation in BIM of a real-world construction project and the benefit section contains the expected applications and advantages. The conclusions and discussion section presents the conclusion, limitations, and suggestion for future studies.

## 2. Literature Review

### 2.1. Automated Construction Analysis Using 4D BIM

In the past, construction scheduling and planning was conducted using the Critical Path Method (CPM) and the Computer-Aided Drawing (CAD). Despite wide uses of the CPM and CPM-based scheduling techniques (Kim et al. 2015; Senouci and Hassan 2008), these approaches have a critical limitation that is the absence of visualization of construction processes. BIM incorporated CPM activities into 3D building objects so that expected construction progresses can be visualized in a timeline of a construction schedule. In many studies, 4D BIM demonstrated its usefulness in construction planning [4], jobsite safety analysis [5], and constructability checking [6].

Taking advantage of rich digital information in BIM, diverse research studies presented methods to automatically conduct several types of construction analyses. Akinci et al. [7] created various types of spaces in 4D BIM for automated workspace conflict identification. Jongeling et al. [8] integrated spatial workflows of multiple crews in a construction project to quantitatively analyze potential productivity losses due to proximity among them. Zhang et al. [3] presented a rule-based safety hazard identification system focusing on falling hazards. Kim and Teizer [9] automatically generated scaffolding objects in 4D BIM, Kim et al. [2] identified potential safety hazards related to the scaffolding in 4D BIM, and the automated safety hazard checking tool was used to assist in decision making [10,11].

### 2.2. Automated Generation of Moving Paths in BIM

In terms of algorithms for path planning and egress path finding, diverse approaches to predicting and analyzing the conditions of available paths and spaces on a construction site have been studied and developed. Soltani et al. [1] presented the applications of path planning algorithms (Dijkstra, A*, and Genetic Algorithms) to generate optimized paths in terms of travel distance, safety risks, and visibility. On a grid-based site representation, three additional layers of visibility, hazard, and distance were incorporated and the weighted-sum of multi-criteria scores was used for optimization. While this research successfully incorporated critical issues (safety and productivity) into path planning, it has limitations: (1) Dynamically changing spatial-temporal conditions cannot be analyzed. (2) Unsafe locations need to be specified manually. Unsafe locations appear and disappear during construction. Considering the two limitations above, the implementation of the proposed approach may require excessive user-defined inputs. One research used room boundaries to define spaces and implement the analysis relying on architectural information [12]. However, it may lead to wrong results because site configuration dynamically changes upon installation and the dismantlement of components and rooms may not exist until a certain time. In terms of safe egress time research, one paper contains a Medial Axis Transform (MAT)-based granular evacuation modeling framework [13] that describes a safe egress time and the density-based evacuation model. Using Voronoi tessellation of a set of points, this study represents paths and its attributes such as widths and areas. The implementation utilizes the geometry and topology of Industry Foundation Classes (IFC) BIM models. Instead of using MAT or Straight Medial Axis Transform (S-MAT) that define a geometric network, this paper using Voronoi tessellation of a set of points could implement density-based analysis such as flow analysis and path widths and

areas. When it comes to the geometry checking applications of a BIM model, various commercial and open source applications have been developed to support the automated validation. The Solibri Model Checker® (SMC), which is a java-based BIM application, verifies an IFC instance file regarding rule sets defined by a user [14]. This commercial application supports diverse types of geometry checking such as object existence, space relations, circulation, fire code exits, path distance checking, and space program checking [15]. Since users can manipulate rule templates embedded in SMC, the IFC instance model can be flexibly validated according to user-defined rules [16]. One research paper also applied the spatial queries of a relational database for evaluating BIM models [15]. In addition, diverse types of BIM-based validation and their challenges regarding the accuracy and interoperability of BIM data have been studied using semantic modular and automated rule-based checkin [17–20]. One paper also demonstrates rule logic according to diverse BIM data validation scenarios according to BIM data exchange standards [21]. In the area of evacuation analyses, Wang et al. [22] and Rüppel et al. [23] illustrate the evacuation path finding using BIM models and virtual reality systems. These papers also describe A* and Dijkstra algorithms for analyzing an egress path, but do not address the interactive evacuation analysis using a 4D BIM model that represents different site conditions according to construction phases.

The investigated research and papers indicate that even though 4D BIM is broadly used, evacuation path planning has not been properly incorporated into 4D BIM. In addition, diverse pathfinding algorithms that have been developed give an insight for developing an interactive evacuation path planning using 4D BIM as a platform.

## 3. Objective and Scope

The primary objective of this paper is to investigate the frameworks for generating and visualizing evacuation plans in 4D BIM and to evaluate its feasibility and practicability throughout a case study. Evacuation path planning needs to take into account various factors associated to available exits, construction activities, equipment, material stacks, and others. Since these objects cannot be easily defined in the early planning stages, these factors should be generated as user-defined parameters according to the particular project requirements and site conditions. With the assumption that these user-input factors are correctly defined, this paper presents the framework and development of the evacuation planning using 4D BIM. For evaluating the accessible and secured evacuation path, this framework adopts the A* algorithm, which is one of path finding and graph traversal methods calculating the shortest and directed path among multiple nodes. Even though there are three path planning algorithms including A*, Dijkstra, and Genetic Algorithm, the A* algorithm can generate more logical and optimal paths than the other two algorithms [1]. Therefore, this paper implements the A* algorithm on top of the 4D BIM platform to identify an evacuation path using project information and user inputs. To assess its accuracy and feasibility, the case study of the real construction project using its BIM model for evacuation path planning is included in this paper. Out of the entire construction period, four critical construction phases were selected to provide the associated project's 4D BIM and user-defined parameters for the pathfinding implementation.

The application of the accurate pathfinding algorithm using user-defined parameters to 4D BIM is critical because daily-changing construction work and site conditions should be evaluated by a robust pathfinding method to generate the daily egress paths. The analyzed daily evacuation paths and secured space information are expected to be shared in daily pre-task tool-box meetings with work crews so they can recognize the accessible and secured pathways to quickly get out of the emergent or dangerous areas. In addition, the analyzed results can be significantly useful to dispatch a rescue team to the right place throughout a secured pathway in the case of construction accidents or emergency situations.

## 4. Framework and Algorithms for Automated Generation of Evacuation Paths in 4D BIM

This section presents a framework and implementation of the proposed evacuation path planning in 4D BIM.

### 4.1. Proposed Framework

Figure 1 illustrates the proposed framework comprising five steps that are "updating 4D BIM", "preparing user input", "generating evacuation paths", "reviewing and selecting paths", and "distributing the information in selected paths to related crews".

**Step 1: Update 4D BIM:** An updated 4D BIM is an essential system input that presents current progresses and configuration of the construction site. There have been research studies (Mani et al. 2009) to track construction progresses automatically and reflect the results in 4D BIM. However, automated progress tracking is out of the scope of this research. This research assumes that the input 4D BIM properly reflects construction progresses, configuration, and shape of a current construction site. Therefore, users of this system should manually track current construction progresses and reflect them in 4D BIM.

**Step 2: Prepare user input:** Using the case study of a real construction project, the authors identified key parameters that can impact the calculation of possible pathways within the interactive BIM-based path planning system. The parameters are used to develop rule-checking features for finding out the shortest path and a shelter within a building or a construction site. Even though the listed variables are not the formalized criteria or standardized requirements for calculating evacuation paths, the variables identified from a real construction project are essential for implementing the proposed 4D BIM-based evacuation planning. The list of parameters will be a great initiator that can establish a generalized list of variables in future research that must be considered to assess evacuation paths using 4D BIM. However, because of varied processes and requirements of construction projects, the formalization of such variables and key factors impacting pathfinding were not conducted in this research. The considerable parameters of evacuation planning that the authors identified from the case study involve the following factors:

- Exits that allow work crews to escape from a construction structure and a jobsite
- Secured spaces that can prevent structure collapse, fire, or flammable gas
- Conditions of paths such as accessible or movable paths (e.g., A rebar work location is not an ideal pathway)
- Workspaces of onsite work crews that can impact site congestion and the flows
- Existing structures and objects that can block the pathway of workers
- Interference of workspaces that are crosslinking planned
- Stacked materials that are supposed to be instantly used for construction works
- Various types of equipment being used near construction tasks
- Temporary structures such as scaffolding

The information about construction activities, equipment, materials, and site conditions are generally documented by a field manager or a superintendent on a daily basis to monitor work progresses. This research was conducted with the assumption that such information can be added as user-defined parameters on 4D BIM models to represent the most recent construction site conditions.

**Step 3: Generate all available evacuation paths:** Based on the two system inputs prepared in step 1 and step 2, the BIM-based path planning system automatically identifies and quantitatively evaluates all available evacuation paths between work locations and exists. In this research, to identify the shortest evacuation path, a distance was considered as a primary criterion.

**Step 4: Review and select evacuation paths:** In this step, superintendents and crews' qualitative reviews automatically generated evacuation paths and selected the paths to be used during construction.

Both the path visualization in 4D BIM and the quantitative evaluation of all available evacuation paths assist in the decision making process for the most accessible and secure path. To account for possible spatial conflicts and situations not reflected in 4D BIM, it is desired for multiple crews and managers to review and select evacuation paths together as part of their pre-task planning.

**Step 5: Distribute selected paths to crews:** After the evacuation paths are selected, the information can be communicated with related work crews in various channels, including printed papers or hand-held computers. Using wireless communication, any changes regarding evaluation paths or other safety relevant issues can be immediately distributed to all the work crews and field managers.

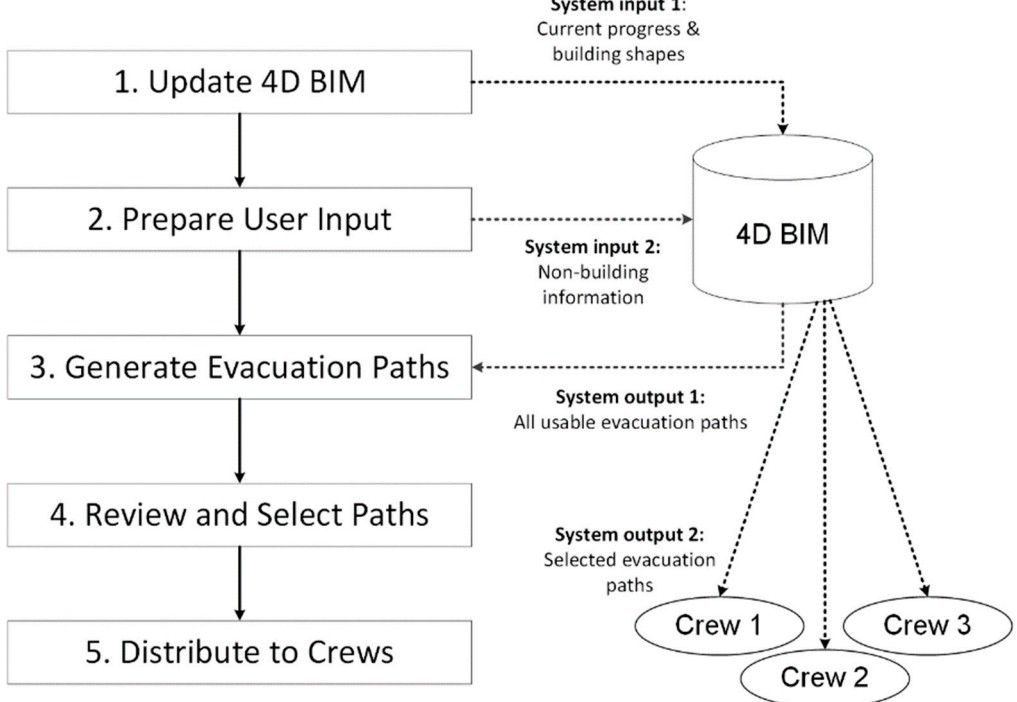

**Figure 1.** Framework for evacuation path planning in 4D BIM.

*4.2. System and Algorithm Development*

4.2.1. Custom 4D BIM Platform for Path Planning

To implement the path finding algorithm, this research employed the 4D BIM platform that the authors customized to import information extracted from Autodesk Revit and manipulate it according to a pre-designated schedule and construction working conditions (Figure 2). A plug-in was developed using Revit Application Programming Interface (API) to automatically extract necessary geometric and non-geometric information from a building model created in Revit. The developed custom 4D BIM platform reads the output XML files for path planning. The custom platform has a user-interface for schedule integration and site component creation.

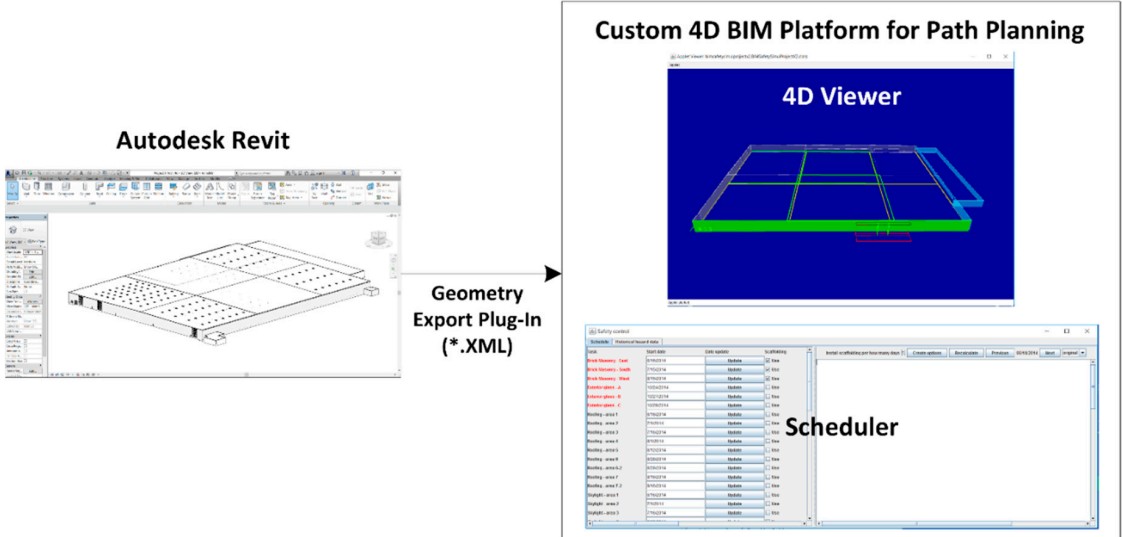

**Figure 2.** Custom 4D BIM platform for proposed path planning integration.

### 4.2.2. A* Searching Application for Path Finding

This study utilizes the A* search algorithm to calculate the shortest path from workers' locations to secured exits in a construction site. Associated job spaces were defined and searched for by drawing a rectangle grid mapped on the entire area of the job site, with 10 feet of node spacing. For each BIM model of four scenarios, the grid nodes were configured according to the distinct site conditions of the particular work day. The nodes in the grid consist of four types: A start node, an exit node, a blocked node (non-navigable), and a regular (non-blocked and navigable) node. Start nodes and exit nodes denote the workers' and exit locations, respectively. Blocked nodes correspond to inaccessible regions such as within 10 ft of installed columns, the inside areas of the steel stack and rebar stack, or areas directly underneath ongoing roofing activities including a scaffolding structure. Non-blocked nodes correspond to all other accessible regions.

Based on the navigation grid, A* path planning finds an optimal path between a start node and an exit node that passes through non-blocked nodes and avoids blocked nodes. It starts with a single node (a start node) and keeps expanding the path by adding nodes until it generates a complete path to the exit node. As shown in Figure 3, 16 immediate neighbors of the current node (as shown in Figure 3) were considered to expand the path.

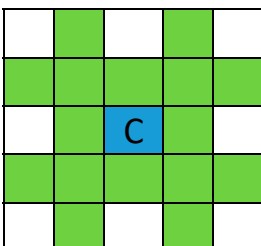

**Figure 3.** A grid node with neighbors.

From the 16 neighbors, blocked nodes and previously explored nodes are removed and the rest of the nodes are evaluated based on the heuristic function F(n).

$$F(n) = G(n) + H(n)$$

where G(n) = distance from start node to node n, H(n) = straight distance from node n to exit node.

## 5. A Case Study and Implementation

To investigate the feasibility and applicability of evacuation path planning, this authors utilized the BIM model of the real construction project (Figure 4a) and implemented the proposed process shown in Figure 1.

### 5.1. Step 1: Update 4D BIM

The first step updated the 4D BIM according to the current status of the project. Due to the large amount of missing information in BIM prepared by the general contractor, four sub-BIM models were extracted and work conditions according to a project schedule were created in coordination with the general contractor. Figure 5 shows the entire schedule of the facility construction involving the work processes of a foundation, a structure, a skin, Mechanical Electrical and Plumbing (MEP), and a roof.

### 5.2. Step 2: Prepare User Input

The second step is to generate non-building components, such as workspaces, temporary structures, and exits. Four milestones that represent the status and progress of each construction activity in the associated domains were defined and used to generate the four sub-BIM models. For the easy organization of work processes and orders, the general contractor of this construction project split the work zone into seven areas (Figure 4b). For example, Scenario Model 1 involves building and site conditions of construction on May 20th 2014, which illustrate foundation work in area 3 and steel structure work in area 2 and 3. The construction work activities and site conditions on May 20th are diagrammed in Figure 6. Each sub-BIM model contains information and its properties pertaining to a work crew location, a work zone, a material stack, a completed structure, an exit, and other factors that can impact generation of evacuation paths.

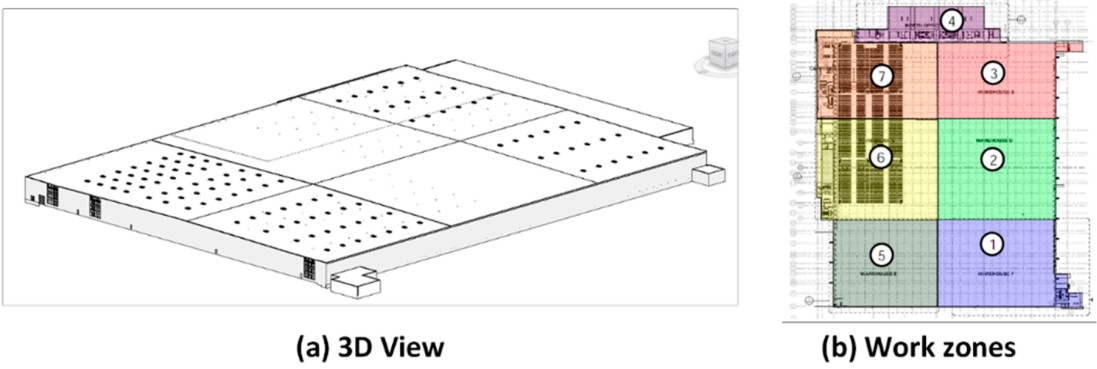

**Figure 4.** (**a**) BIM model and (**b**) work zones of case study project.

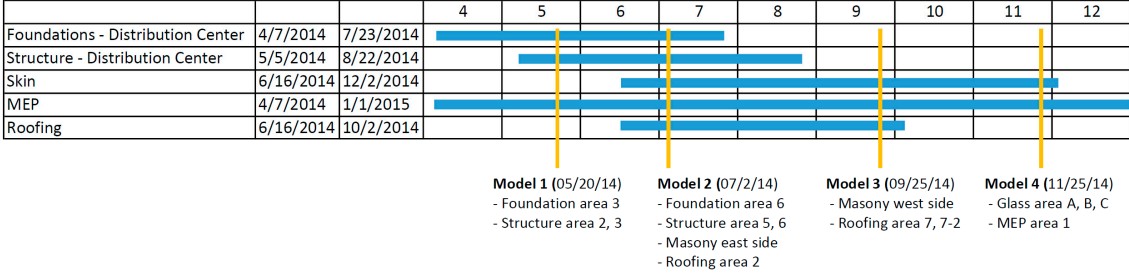

**Figure 5.** Construction schedule and four milestones.

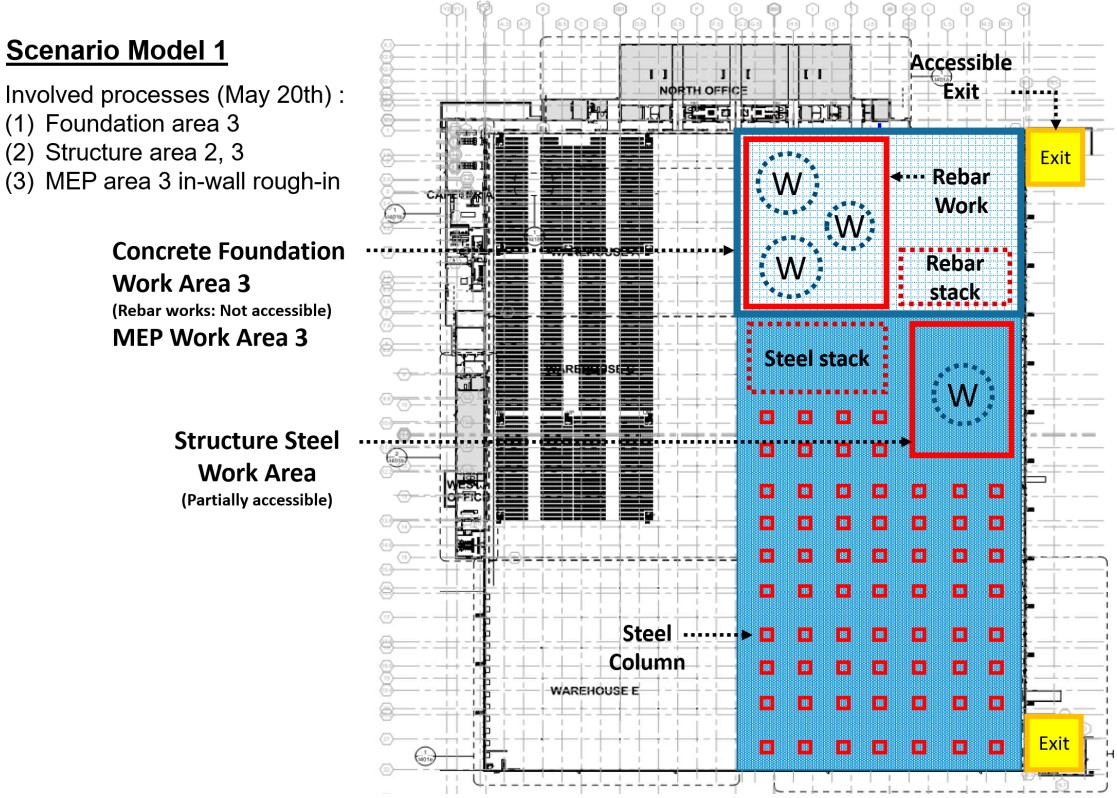

**Figure 6.** Scenario Model 1: Work status and site conditions.

Scenario Model 1 shows the following three working processes: (1) Foundation work in the area 3, (2) structure work in area 2 and 3, and (3) MEP work in area 3. Each domain activity is color-coded on the floor plan to efficiently represent the types of construction work. On May 20th, area 3, having a concrete foundation task with a reinforcing bar (rebar) installation, is only accessible from the nearest exit in yellow. In other words, rebar workers can use the exit in the upper side of the floor plan, but structure steel workers cannot use it for the evacuation path. Thus, the accessible exit for steel workers is the one located in the bottom of the plan. In addition, each work activity entails associated materials. The location and the area of materials such as a rebar stack is an imperative component for evaluating real-time evaluation planning. However, this manuscript does not involve any formularized and generalized information about work types, areas, and considerable parameters because construction projects generally involve significantly distinct and unique project types, program requirements, tasks, and a schedule that result in different site conditions. For the real-time evacuation analysis, this information can be manually tailored and updated by a BIM modeler according to a project and site information.

Scenario Model 2 shows the following five working processes (Figure 7): (1) Foundation work in area 6, (2) structure work in area 5 and 6, and (3) masonry work in the east side, (4) MEP work in area 1 and 6, and (5) roofing work in area 2. On July 2nd, the roofing work area containing scaffolding cannot be used as an accessible evacuation pathway because of the shallow path spaces, the complex scaffolding structure, and the low illumination level of the area.

Scenario Model 3 in Figure 8 shows the following three working processes: (1) Masonry works in the west side, (2) MEP work in area 3 and 5, and (3) roofing work in area 7 and 7-2.

Scenario Model 4 in Figure 9 shows the following two working processes: (1) Masonry work in the east and west sides and (2) MEP work in area 1.

## Scenario Model 2

Involved processes (July 2nd) :
(1) Foundation area 6
(2) Structure area 5, 6
(3) Skin (East side: masonry)
(4) MEP area 1 rough-in, area 6
     in-wall rough-in
(5) Roofing area 2

**Concrete Foundation
Work Area 6**
(Rebar works: Not accessible)
**MEP Work Area 6**

**Structure Steel**
(Partially accessible)

**MEP Work Area 1**

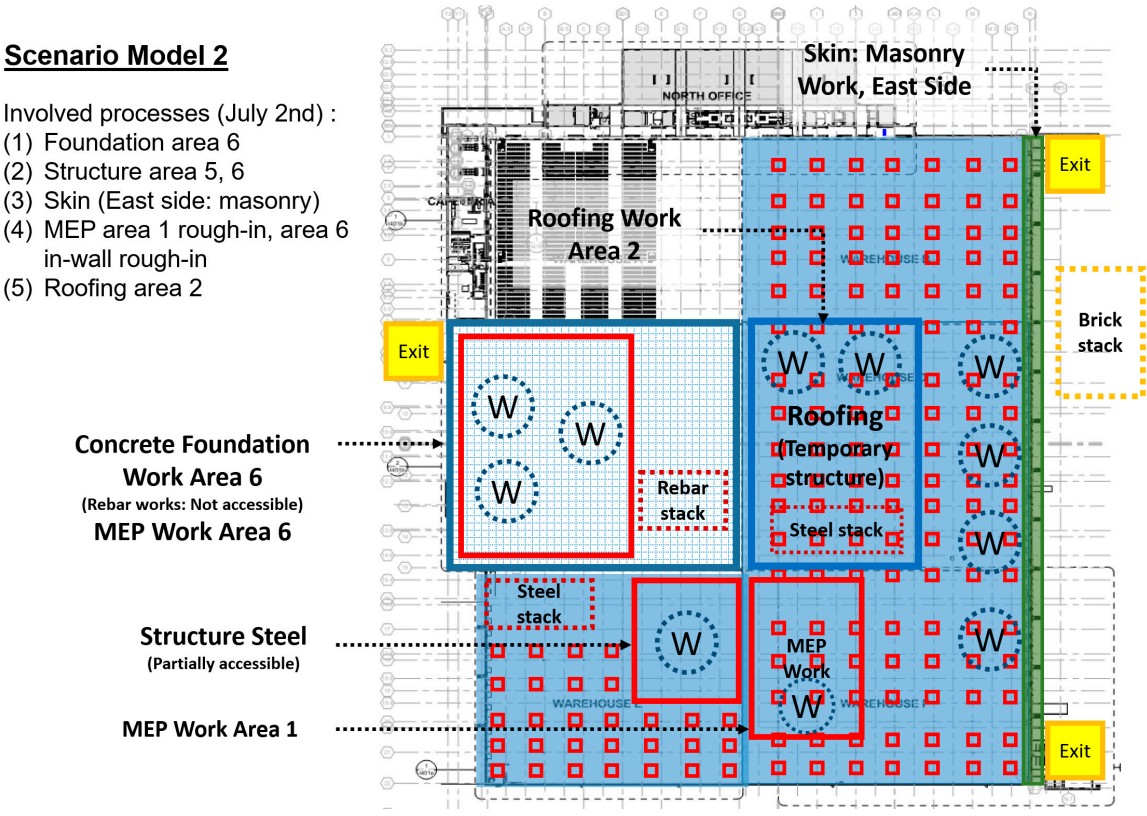

**Figure 7.** Scenario Model 2: Work status and site conditions.

## Scenario Model 3

Involved processes
(September 25th) :
(1) Skin (West side: masonry,
     area B Fabricate Glass)
(2) MEP area 3 rough-in, area 5
     rough-in
(3) Roofing area 7, 7-2

**Roofing Work
Area 7, 7-2**

**MEP Work
Area 5**

**Skin: Masonry
Work, West Side**

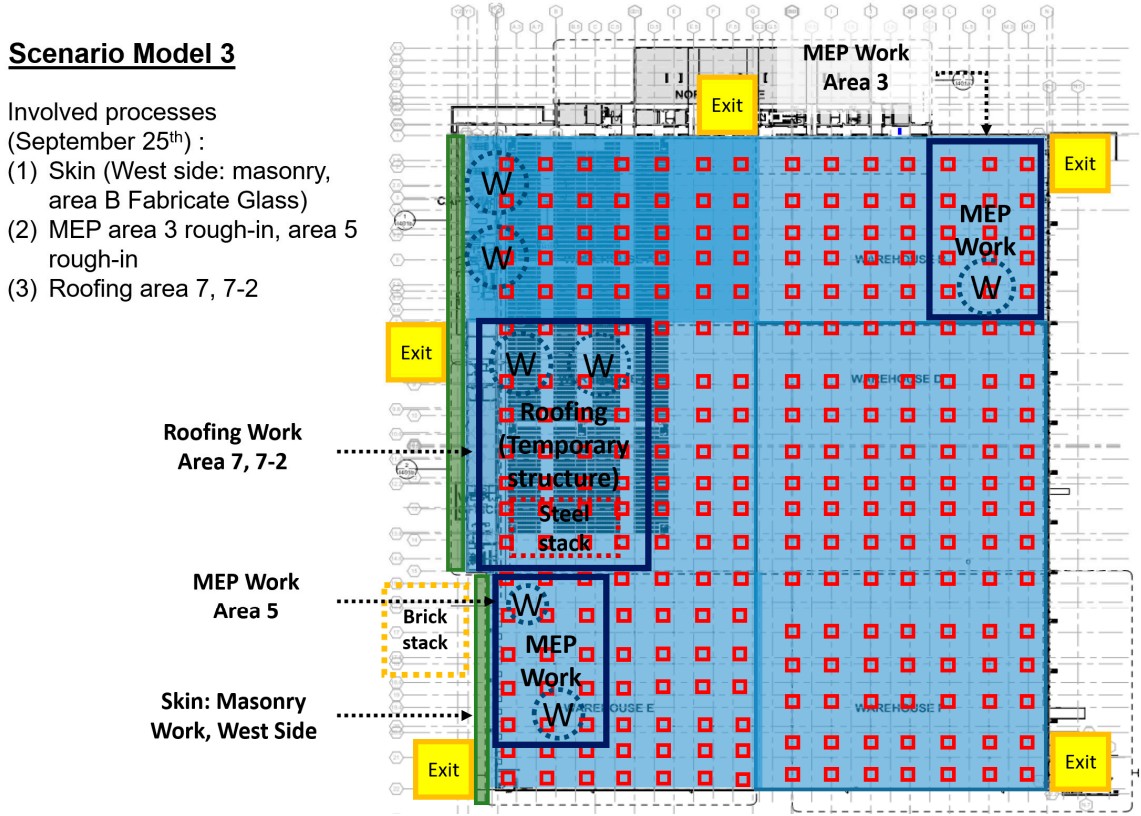

**Figure 8.** Scenario Model 3: Work status and site conditions.

**Scenario Model 4**

Involved processes (November 25th) :
(1)  Skin (East, west side: door, area A, B, C: glass )
(2)  MEP area 1 rough-in, interior office center

MEP Work Area 1

**Figure 9.** Scenario Model 4: Work status and site conditions.

*5.3. Step 3 and 4: Generate All Available Evacuation Paths and Select Paths for Distribution*

As shown in the four scenarios, work areas, task types, and accessible exits are varied according to construction progress, a schedule, a weather, a logistics, site conditions, or diverse variables. To secure evacuation paths of a changing construction site, automated and dynamic path planning and identification are required for generating a daily evacuation plan. This paragraph illustrates the detailed implementation and its processes. The construction site environment and work activities related to the four key scenarios in Section 5.1 were reflected in the developed path planning platform. Then, this system is designed to automatically identify and evaluate accessible evacuation paths. Figure 10, Figure 15, Figure 16, and Figure 17 present the results of the four scenarios in the four different views (site conditions presented in 4D BIM, a navigation grid, all available paths, and selected paths).

Figure 11 illustrates the site condition of scenario 1 including work locations, structural columns, exits, etc. As shown in Figure 12, a navigation grid of 10 feet granularity was created. The navigation grid presents accessible and inaccessible places within the construction site. Based on the site conditions (Figure 11) and a grid (Figure 12), all available evacuation paths were identified and evaluated in terms of a total distance (Figure 13). Finally, one or two evacuation paths were selected by users as shown in Figure 14.

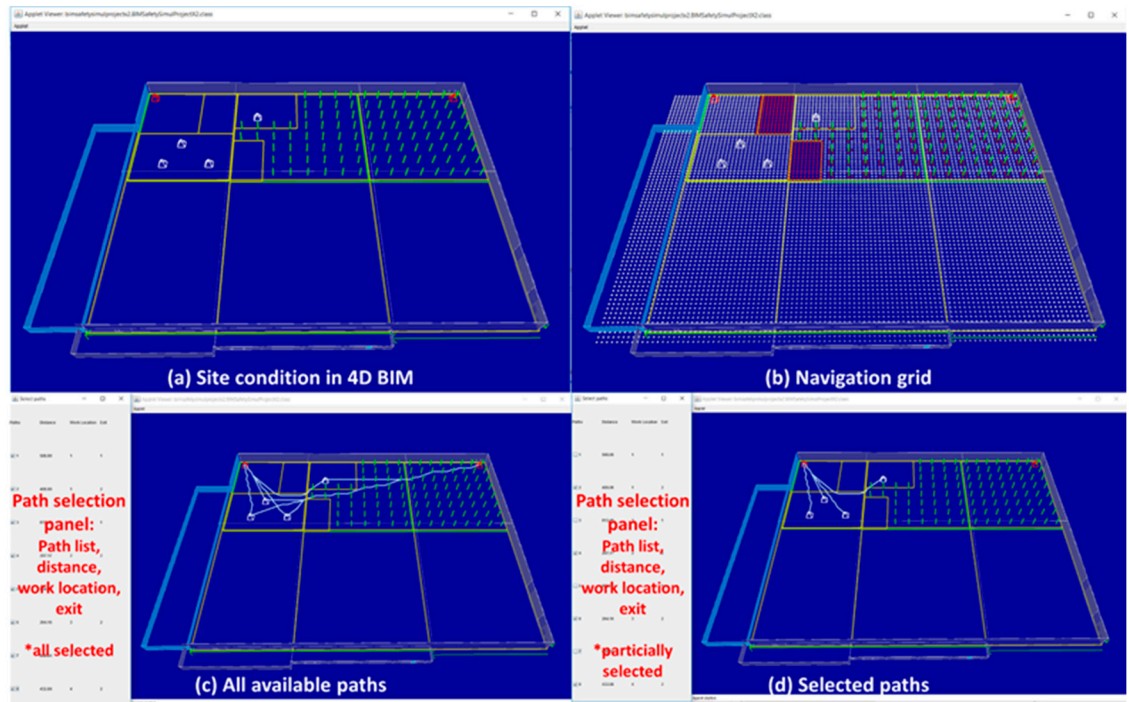

**Figure 10.** Scenario 1 (**a**) site condition, (**b**) navigation grid, (**c**) all available paths, and (**d**) selected paths.

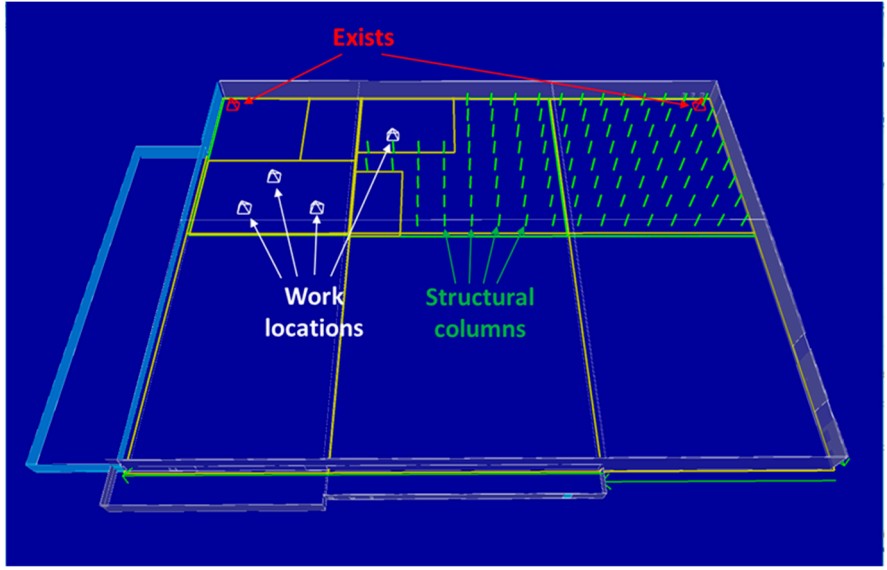

**Figure 11.** Site condition of scenario 1 reflected in the prototype system.

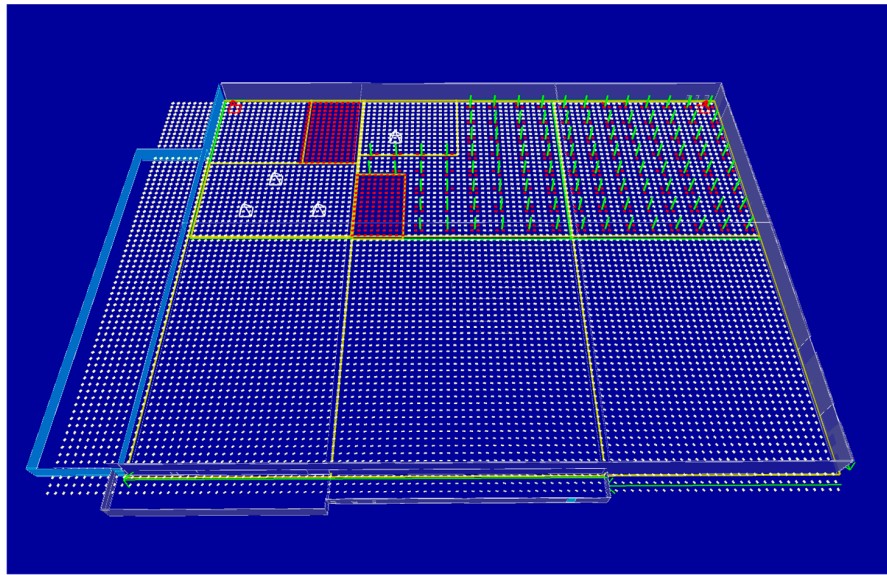

**Figure 12.** Scenario 1 navigation grid with accessible and inaccessible nodes.

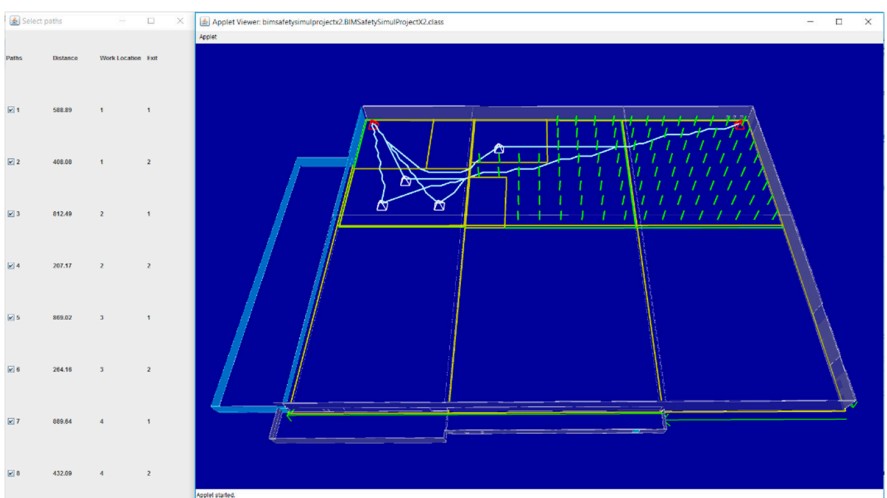

**Figure 13.** All evacuation paths found from scenario 1.

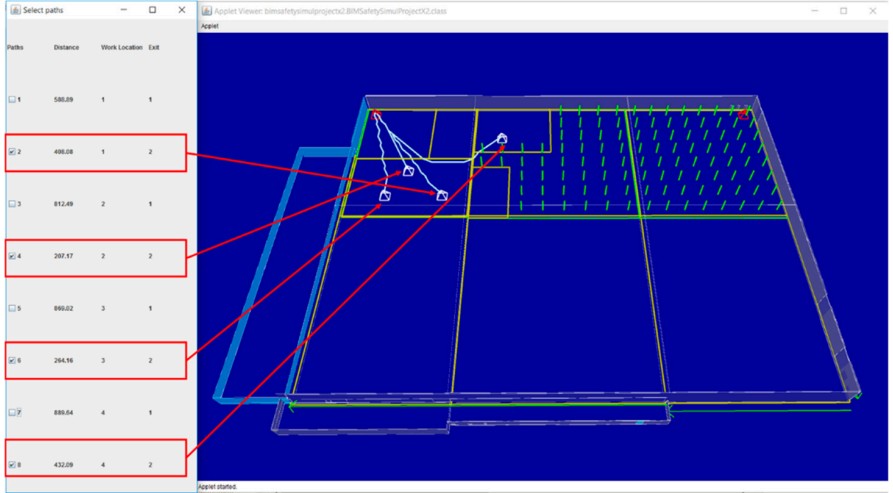

**Figure 14.** Selected evacuation paths found from scenario 1.

Figure 15, Figure 16, and Figure 17 show the results from scenario 2, 3, and 4, respectively. Figure 18 shows paths from roofing work locations. This case study assumed that crews working for roofing generally go down to the first floor level through the opening in the middle of the roofing area. For the four scenarios, the developed system successfully generated rational evacuation paths for each work location by properly considering the construction site conditions and explicitly represent secured circulation paths connecting to the exits. Even though final decisions on determining the shortest path can also be made automatically, based on calculated distances, the proposed approach puts its objective to assist the decision making process of domain professionals with the automatically generated pathways.

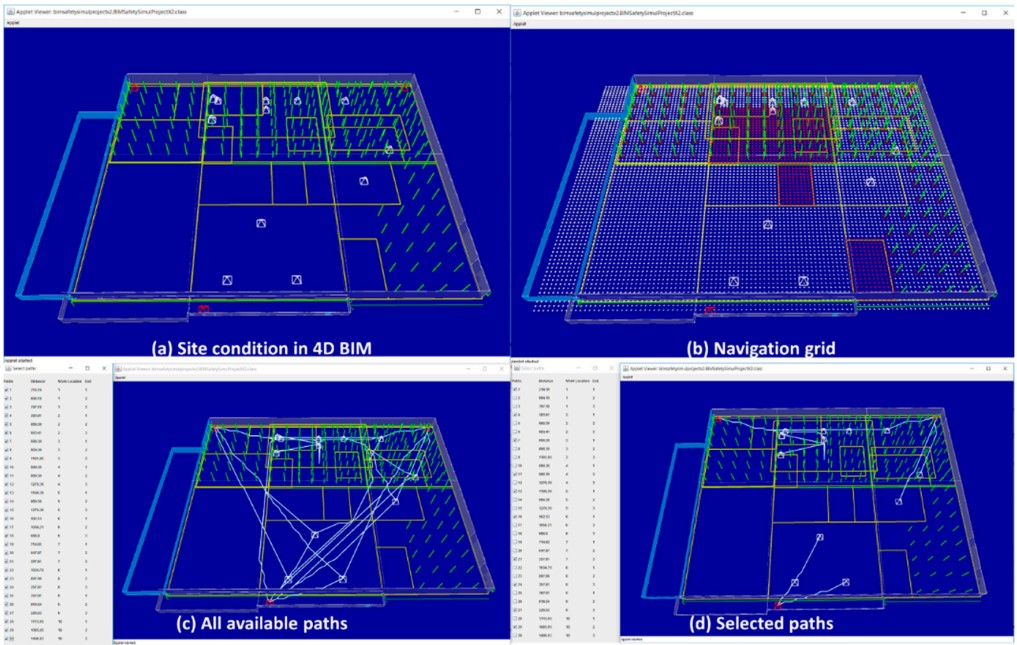

**Figure 15.** Scenario 2 (**a**) site condition, (**b**) navigation grid, (**c**) all available paths, and (**d**) selected paths.

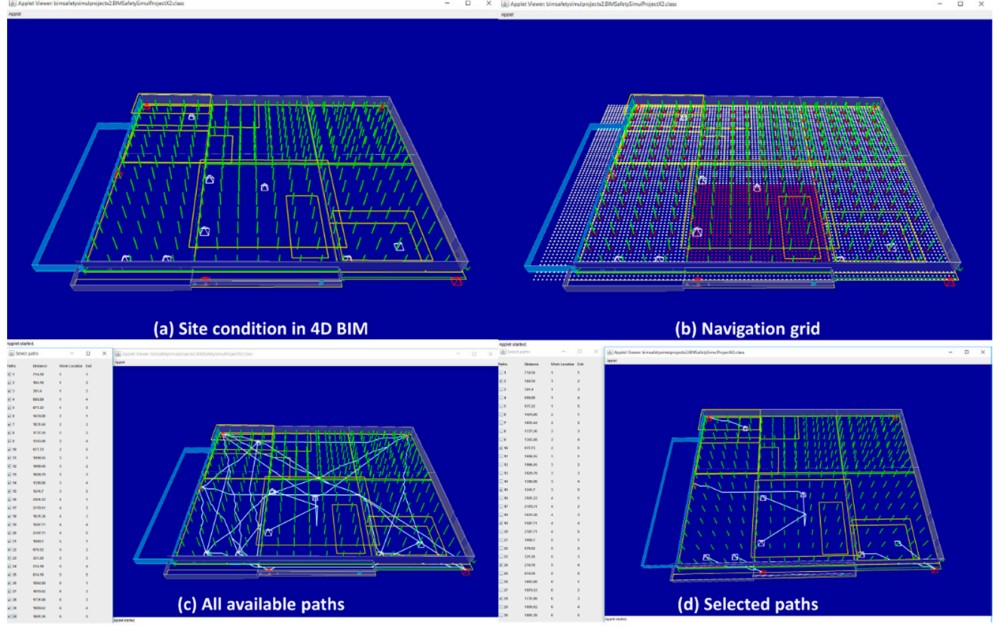

**Figure 16.** Scenario 3 (**a**) site condition, (**b**) navigation grid, (**c**) all available paths, and (**d**) selected paths.

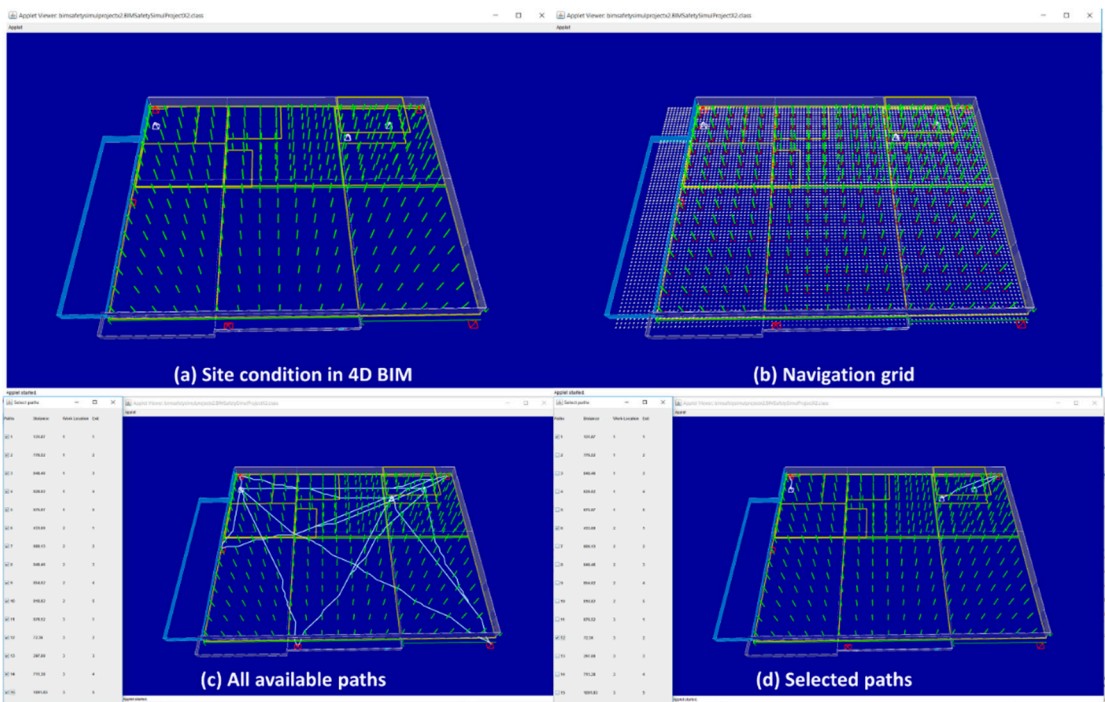

**Figure 17.** Scenario 4 (**a**) site condition, (**b**) navigation grid, (**c**) all available paths, and (**d**) selected paths.

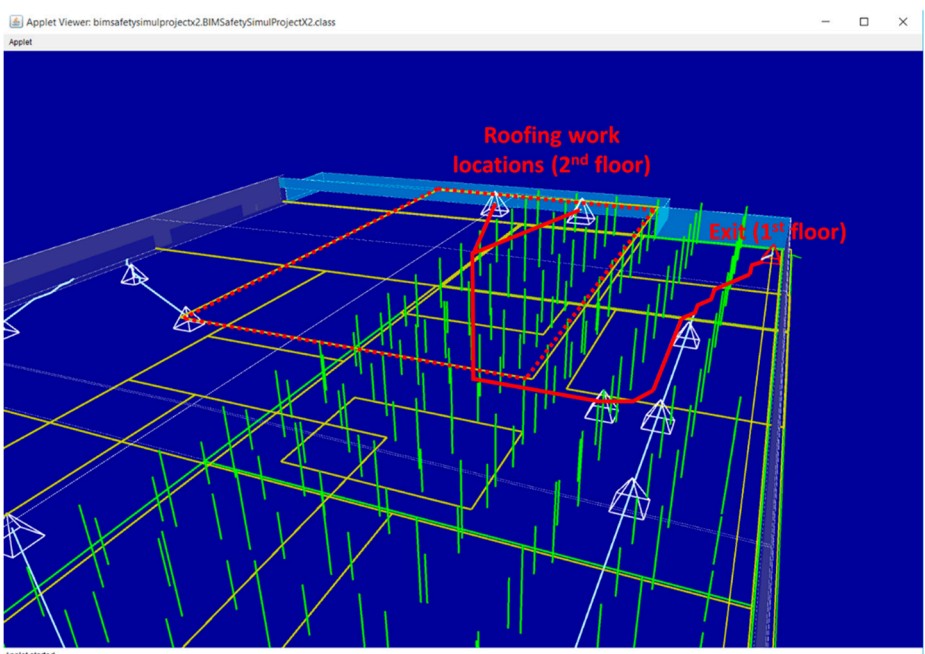

**Figure 18.** Evaluation paths from roofing work locations in scenario 2.

*5.4. Step 5: Distribute Selected Paths to Crews*

After generating all the evacuation paths of the four project milestones, certain evaluation paths were selected based on the project-specific review as discussed in Section 5.3. The research team provided information on the selected evacuation paths to the general contractor for distribution to the work crews of the project. Although there was no incident of safety hazards that required evacuation from the construction site, the general contractor was able to provide evacuation paths that were generated based on BIM containing up-to-date project status as well as construction site conditions that are impacted by non-building components, such as workspaces, temporary structures, etc.

*5.5. Benefits and Potentials*

The proposed approach, which uses the customized 4D BIM platform, has potential for any commercial 4D BIM tool such as Synchro or Vico. The pathfinding algorithm and process can be integrated into any 4D BIM application to generate egress paths and circulation paths. The key benefit of 4D BIM-based path planning is the automated generation of daily, weekly, or monthly updatable and changeable evacuation plans that reflect project progress, planned work schedules, and site conditions. Users can define work zones, schedules, work orders, and other required parameters on the 4D BIM platform to flexibly manage BIM models and analyze possible pathways. The accessible pathways, secured spaces, and installed exits shown in 4D project models that can be shared and trained in every day's tool-box meetings, will enhance emergency preparedness planning and help a rapid reaction in any emergency situations or construction accidents. In particular, in emergency situations, such as a collapse or fire on site, a dispatched rescue team can quickly recognize the site conditions using the mapped 4D models and determine the shortest and secured pathway to save labors from the site. From a project management perspective, the daily updated geometric representation of work zones and a construction site will improve logistics planning, task sequence management, and material stack/equipment movement planning by incorporating path planning and visualization capabilities into 4D BIM and daily pre-task planning. Since a construction project has to organize a series of tasks, numerous labors, diverse materials within a limited jobsite space, a site logistic, and work planning, using 4D BIM and its automated circulation path finding features will play a pivotal role in the project management of diverse construction projects. Furthermore, an explicit illustration of the existing conditions of project tasks and the available movement paths of equipment or material, will facilitate more effective construction project management and planning.

## 6. Discussion and Conclusions

Establishing evacuation paths for multiple work crews based on changing construction site conditions requires a significant amount of manual and labor-intensive tasks. Most of construction projects do not provide situation-specific evacuation plans for workers through the construction. To resolve this challenge, this research proposed a framework to automatically generate available evacuation paths for multiple crews using project information mapped in 4D BIM. To account for construction site components, such as workspaces, exists, and temporary structures, that are difficult to model in 4D BIM in advance, the developed prototype system requires users to manually create these components as part of a daily pre-task planning. A case study on four real-world construction scenarios successfully generated rational evacuation paths for multiple work crews.

Even though this research shows valuable results pertaining to the dynamic and automated evacuation path planning, there are multiple limitations that should be addressed. (1) The first limitation is a manual 4D BIM update. Since the proposed approach uses current geometric conditions as an essential system input, 4D BIM should reflect the construction site's conditions accurately. However, identifying construction progresses and reflecting the current status is a tedious process. Due to this importance, many research studies attempted to automate the process of updating BIM and its embedded information. However, this is out of the scope of the research presented in this paper. (2) The second limitation is a manual user input for non-building components in construction sites. Other than building geometric information, non-building elements, such as workspaces, temporary structures, major tools, storage areas, need to be identified and modeled in 4D BIM environment by the users. To achieve this, construction, VDC, or BIM managers of a construction project should coordinate with work crews and specify their work locations accurately in the system. (3) The third limitation is the reliance for situation-specific heuristic functions. Depending on the project sizes, spatiotemporal conditions, and preferences of users, different search objectives and heuristic functions should be designed. In the case study of this research, we used the total moving distance as the path finding objective. However, for construction projects with different conditions and requirements, objectives and heuristic functions should be designed according to project-specific conditions. In order to address

these challenges, future research should involve the investigation of the key factors that should be taken into account by the implementation framework of evacuation path finding. In addition, a well-structured user interface for developing a 4D BIM and manipulating site conditions is required for accurately identifying accessible paths and practically adopting research findings to construction projects. The mobile-based approach to share and disseminate real-time evacuation paths to labors and rescue teams will be further studied to bolster this egress path planning research and construction safety environment.

**Author Contributions:** Conceptualization, K.K. and Y.-C.L.; software development, K.K.; validation, Y.-C.L.; original draft preparation, K.K. and Y.-C.L.

**Funding:** This research received no external funding.

**Acknowledgments:** The authors appreciate Sajiva Pradhan a research assistant at University of Houston for her assistance in implementing A* searching algorithm during the case study.

**Conflicts of Interest:** The authors declare no conflict of interest.

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
