# Peer review of "Automated Generation of Daily Evacuation Paths in 4D BIM"

_applsci, doi:10.3390/app9091789_

Round 1
Reviewer 1 Report
A well-written and original paper.
The topic is obviously of importance.
My only comment would be that some of the figures are small, not very clear and should be improved so that the reader can understand what they contain.
Other than that, a very good paper.
Author Response
Thank you for your positive comments on our work.
Reviewer 2 Report
The authors discuss an interesting and important topic in the construction domain. However, the reviewer thinks key research information was not clearly demonstrated and explained in the manuscript. The following are the reviewer’s comments for the authors’ reference.
1. The authors highlight that the proposed approach considers the dynamic feature of a construction site, but in the illustration of the four scenarios of the case study, nothing reflecting the dynamic feature was actually discussed and analyzed. The authors only show evacuation paths from certain fixed points in the scenarios. Workers on a jobsite in fact move frequently and thus, their evacuation paths should vary from time to time. No details about this issue was discussed in the manuscript.
2. The reviewer has another major concern about the potential application of the proposed approach in reality. Specifically, the reviewer cannot see concrete superiority of the proposed approach over traditional methods, such as workers’ qualitative and imaginative planning in mind and paper-based discussions to think of an evacuation path daily in a tool box meeting. Even when evacuation need to be carried out on site, workers are not possible to check the predetermined path with the system at the moment due to emergency and urgency, especially considering the dynamic feature as mentioned previously.
3. The reviewer agree with the authors that site conditions are changing and each project has its own unique layouts of materials, construction processes, structural components and crews, making evacuation paths different from project to project. However, the reviewer think the authors should generally demonstrate what key factors should be considered when planning evacuation paths in Section 4.1 (Step 2 of the proposed framework), rather than merely picking one case project and analyzing the factors to be considered for the case project. This way can make this research have more contributions to the domain knowledge about evacuation path planning.
4. The reviewer wonders why in the case study only two areas (Area 3 and Area 6) requires foundation work. Clear introduction of the case is necessary for readers to understand the scenarios.
5. In the implementation result of scenario 1, all selected evacuation paths direct to the exit in the upper side of the floor plan, but it conflicts with the rule “structural steel worker cannot use the exit for evacuation path” in line 319. Is there any mistake here?
6. How the A* algorithm was implemented to get rules for generating evacuation paths is not clearly explained in the manuscript. Details are necessary to be supplemented in Section 4.2.2 and in the case study.
7. Duplicated sentences are found in lines 135-136.
8. There are certain English grammatical mistakes and please check the manuscript thoroughly.
Author Response
The detailed answers were added in the attached file.

Reviewer 3 Report
This paper addresses an important research area of utilizing BIM models for construction safety. The research objective and scope, literature review, and research method were well defined and illustrated. However, it is recommended to re-organized some part of the main text in order to make this draft more interesting.
- The proposed framework in the Section 4.1 and Figure 1 has four steps for evacuating path planning in 4D BIM. However, The Section 4.2 (Systems and Algorithms) and Section 5 (Case Study) did not clearly elaborate how each step is designed and implemented. It would help to follow the ‘five steps’ in Figure 1 for these Sections. It is also recommended to add some practical implications as well as contributions from this paper based on the five-step framework.
- Minor corrections required.
Line 134: Medial Axis Transform (MAT) should be spelled out in the line 134, not in 138
Line 188: Nee to be corrected to “Section 4.1”
Author Response

(The authors gave the same response as above.)
